# Lung and blood early biomarkers for host-directed tuberculosis therapies: Secondary outcome measures from a randomized controlled trial

**Robert S. Wallis**[1,2,3]*, **Sibuse Ginindza**[1], **Trevor Beattie**[1,4], **Nishanee Arjun**[1], **Morongwe Likoti**[1], **Modulakgotla Sebe**[1], **Vinodh A. Edward**[1,4,5,6], **Mohammed Rassool**[6,7], **Khatija Ahmed**[8], **Katherine Fielding**[9], **Bintou A. Ahidjo**[1], **Mboyo D. T. Vangu**[6], **Gavin Churchyard**[1,9]

1 Aurum Institute, Johannesburg, South Africa, 2 School of Medicine, Vanderbilt University, Nashville, Tennessee, United States of America, 3 School of Medicine, Case Western Reserve University, Cleveland, Ohio, United States of America, 4 Department of Interdisciplinary Social Science, School of Public Health, Utrecht University, Utrecht, The Netherlands, 5 Yale School of Public Health, Yale University, New Haven, Connecticut, United States of America, 6 Schools of Pathology (VAE) and Medicine (MDTV), University of the Witwatersrand, Johannesburg, South Africa, 7 Clinical HIV Research Unit, Johannesburg, South Africa, 8 Setshaba Research Centre, Soshanguve, South Africa, 9 Department of Medical Statistics and Epidemiology, London School of Hygiene and Tropical Medicine, London, United Kingdom

* rwallis@auruminstitute.org

**Data Availability Statement:** Data cannot be made publicly available because participants did not consent to having their data shared publicly. The

## Abstract

### Background

Current tuberculosis treatments leave most patients with bronchiectasis and fibrosis, permanent conditions that impair lung function and increase all-cause post-TB mortality. Host-directed therapies (HDTs) may reduce lung inflammation and hasten eradication of infection. Biomarkers can accelerate tuberculosis regimen development, but no studies have yet examined early biomarkers of TB-HDTs.

### Methods

Biomarkers of inflammation and microbicidal activity were evaluated as a part of a recent phase-2 randomized controlled trial of four HDTs in 200 patients with pulmonary tuberculosis and baseline predictors of poor outcome, including CC-11050 (PDE4i), everolimus (mTORi), auranofin (oral gold salt), and ergocalciferol (vitamin D). Two of the 4 arms (CC-11050 and everolimus) showed superior recovery of lung function at day 180 compared to control; none showed accelerated eradication of MTB infection. Patients underwent [18]F-fluorodeoxyglucose positron emission tomography/computed tomography (PET/CT) on entry and day 56. PET signals were analyzed according to total, maximal, and peak glycolytic activity; CT was analyzed according to total modified Hounsfield units to assess radiodensity. Mycobactericidal activity in *ex vivo* whole blood culture was measured on days 42, 84, and 140. C-reactive protein (CRP) was measured at multiple time points.

consent forms were approved by the University of the Witwatersrand Human Research Ethics Committee. In accord with the data access provisions specified in the grant agreement, however, anonymized patient level data can be made available to qualified researchers through the London School of Hygiene and Tropical Medicine (contact via researchdatamanagement@lshtm.ac. uk or via doi.org/10.17037/DATA.00002277).

**Funding:** The study was supported by grant OPP1127276-2015 from the Bill & Melinda Gates foundation to RSW and ACT4TB/HIV from the South African Medical Research Council to GC, and by grant DRTB-HDT 847465 from Horizon 2020, which was used to support SG's work to prepare the database and dictionary. The funders had no role in study design, data collection and analysis, decision to publish, or preparation of the manuscript.

**Competing interests:** The authors have declared that no competing interests exist.

## Results

All PET/CT parameters showed highly significant reductions from baseline to day 56; however, only maximal or peak glycolytic activity showed further experimental reduction compared to controls, and only in everolimus recipients. CRP dropped precipitously during early treatment, but did so equally in all arms; over the entire period of treatment, the rate of decline of CRP tended to be greater in CC-11050 recipients than in controls but this fell short of statistical significance. Whole blood mycobactericidal activity in *ex-vivo* culture was enhanced by auranofin compared to controls, but not by other HDTs.

## Conclusions

None of these early biomarkers correctly predicted HDT effects on inflammation or infection across all four experimental arms. Instead, they each appear to show highly specific responses related to HDT mechanisms of action.

## Introduction

Tuberculosis is a leading cause of morbidity and mortality globally [1]. Current treatments require patients adhere closely to multi-drug regimens that are long, complex, and often poorly tolerated or ineffective. Even if cured, most patients are left with bronchiectasis and fibrosis, permanent conditions that impair lung function and increase all-cause mortality [2–5]. The antimicrobial tuberculosis treatments currently in development are unlikely to affect these outcomes.

There is growing interest in the potential role of adjunctive host-directed therapies (HDT) to address these unmet needs. TB-HDTs promise to protect the lung and shorten treatment by reducing lung inflammation, enhancing intralesional drug penetration, and inducing cellular antimicrobial activity [6]. Although many TB-HDT candidates have been proposed based on observations in cell culture or animal models, few have been definitively evaluated in clinical trials.

Biomarkers have the potential to accelerate tuberculosis drug development. Although they may assess a wide range of processes or responses, the most robust biomarkers measure factors that are essential to underlying pathologic processes, as these are presumably best able to capture the full effects of many types of interventions on clinical outcomes [7]. It is not yet clear how the interplay of host and microbial processes will influence the role of biomarkers in the evaluation of TB-HDTs.

We recently published findings for safety and preliminary efficacy of an experimental medicine phase-2 randomized controlled trial (TB-HDT) of four adjunctive host-directed tuberculosis therapies [8]. In our study, patients with moderate or far-advanced pulmonary tuberculosis and heavy sputum burdens of *M tuberculosis* infection were randomly assigned to either rifabutin-substituted short course therapy alone, or that plus one of four HDTs for 4 months. Although none significantly accelerated sputum culture conversion, two candidates (CC-11050, a type 4 phosphodiesterase inhibitor, and everolimus, an mTOR inhibitor), enhanced the recovery of the 1-second forced expiratory volume (FEV1) at month-6. Auranofin (an oral gold salt) and ergocalciferol (vitamin D) were ineffective. FEV1 is both a direct measure of lung function and an independent, generalizable predictor of all-cause mortality, even in individuals without recognized lung disease [3]. The finding of superior recovery of

FEV1 at month-6 (two months after cessation of host-directed treatment) may indicate effects on post-inflammatory airway remodeling, potentially mitigating excess post-TB mortality. However, the late appearance of this potentially important finding (at the end of tuberculosis treatment) limits the use of FEV1 as an early tuberculosis biomarker.

We here report secondary study findings regarding early biomarker responses, including measures of glycolytic activity and radiodensity using [18]F-fluorodeoxyglucose (FDG) positron emission tomography and computed tomography (PET/CT) in the lung, and measures of inflammation and mycobactericidal activity in blood. The objective was to better understand the potential of these early biomarkers to assess TB-HDTs.

## Methods

### Study design and participants

Patients were men and women aged 18–65 years with Cepheid Xpert MTB/RIF sputum testing showing rifampin-susceptible *M tuberculosis*, and ≥1 probe showing a cycle threshold (Ct)< 20; moderately advanced or far advanced pulmonary tuberculosis by chest radiography; body weight 40-90kg; and willing and able to provide written informed consent according to South African and international guidelines prior to any study procedures. Patients with prior tuberculosis, HIV-1 infection, chronic hepatitis B virus infection, diabetes mellitus, those requiring corticosteroids or other prohibited medications, or with chemistry or hematology values outside of specified ranges, were excluded.

The protocol received ethics approval from the University of Witwatersrand Human Research Ethics Committee (reference number 151112) and the London School of Hygiene & Tropical Medicine (reference number 10645), and is registered as study 4297 in the South African Human Research Electronic Application System. It was approved by the South African Medicines Control Council (MCC, now SAHPRA), reference number 20160506. The trial is registered by the South African National Clinical Trial Registration as DOH-27-0616-5297, and by ClinicalTrials.gov as NCT02968927. Study progress was monitored at regular intervals by an independent Data and Safety Monitoring Committee.

### Randomization and blinding

Patients were randomly assigned to: CC-11050 200mg BID with food; everolimus 0.5mg QD; auranofin 6mg QD after an initial week of 3mg QD; ergocalciferol 5mg on day 1, then 2.5mg on days 28 and 56; or control, in equal blocks of 10 with stratification by site, using envelopes provided to each site by the study statistician (SG). CC-11050, everolimus, and auranofin were given from days 1–112. All patients additionally received standard tuberculosis treatment with rifabutin (Rb) 300mg QD substituted for rifampin (2HRbZE/4HRb), due to potentially deleterious pharmacokinetic interactions of everolimus and CC-11050 with rifampin. CC-11050 was provided by Celgene. Laboratory personnel and senior study leadership (RSW and GC) were blinded as to patient treatment assignment until all study treatments were completed.

### Procedures

Patients were recruited at three sites in the greater Johannesburg-Pretoria area: the Tembisa Clinical Research Centre (CRC), Tembisa; the Clinical HIV Research Unit, Johannesburg; and the Setshaba Research Centre, Soshanguve. The full protocol appears online at https://bit.ly/2MA6bJL.

[18]F-FDG PET/CT scans were performed at a single facility (Charlotte Maxeke Johannesburg Academic Hospital, University of the Witwatersrand) on study days 1 and 56. During

winter months, patients received a single dose of propranolol prior to imaging to inhibit glucose uptake by brown fat [9]. Regions of interest (ROIs) were outlined by a single reader (RSW) using MIM image analysis software (mimsoftware.com) to include the lungs but exclude mediastinal and other thoracic structures. The ROIs were first selected using fused transverse sections and reviewed using sagittal sections. The reader was blinded as to treatment assignment. PET/CT parameters were log-transformed to improve normality. PET parameters included total standardized uptake value (SUVbw*ml, the product of body weight adjusted intensity and volume), maximum SUVbw (based on single voxels), and peak SUVbw (based on 1-cm spheres, intended to reduce random noise due to single voxel errors). Radiodensity (CT) was assessed as modified total Hounsfield Units (mHU*ml, the product of modified radiodensity and volume). The HU scale assigns air a value of -1000 and water a value of zero. The scale was modified by adding 1000 to HU values and then dividing by 1000. The resulting positive values fall on a numerical scale similar to SUV.

CRP was measured at a single laboratory (BARC, Johannesburg). Nominal values were used to examine changes from baseline to study day 56. Log-log-transformed values were used to evaluate effects over the full period of treatment, to enhance the linearity of changes over time.

Whole blood bactericidal activity (WBA) was measured on study day 42 (during the intensive phase), day 84 (during the continuation phase), and day 140 (after experimental treatments had ceased). Blood was stored at room temperature with slow constant rotation until a full set of participant samples (0–8 hr) had been collected, at which time they were transported to the Tembisa laboratory for testing. WBA against *M. tuberculosis* H37Rv was determined as previously described [10]. Briefly, *M. tuberculosis* H37Rv was grown in MGIT and frozen in aliquots at -80˚C. A titration experiment determined the relationship between inoculum size and TTP, and identified the volume positive in 5.5 days. Whole blood cultures consisted of 300 µl heparinized blood, an equal volume of RPMI 1640 tissue culture medium (Highveld Biological, Lyndhurst, South Africa), and mycobacteria from the specified volume of stock. After 72 hrs incubation at 37˚C with slow rotation, cells were sedimented, the liquid phase removed, and blood cells disrupted by hypotonic lysis. Bacilli were recovered and inoculated into MGIT and incubated until flagged as positive. Log change in viability was calculated as log(*final*)–log(*initial*), where *final* and *initial* are the volumes corresponding to the TTP of the completed cultures and its inoculum control, respectively, based on the titration curve. Results were expressed as log change per day of whole blood culture, with positive values indicating growth. Cumulative WBA over 24 hrs was calculated as the $AUC_{0-24}$, and expressed as $\Delta$log/d•d, or simply as log change. The full protocol is available online at https://bit.ly/2QEZbAZ.

## Statistical analysis

The intent to treat (ITT) population included all randomized patients who received at least one dose of study drugs. The modified ITT (mITT) population included patients in the ITT population, but excluded those wrongly enrolled (*ie*, not meeting enrollment criteria). The per protocol (PP) population included patients in the mITT population, but excluded those who did not complete treatment or were found to have inadequate adherence based on pill counts indicating <299 rifabutin 150mg tablets (83%) taken in total. The ITT population was specified as the main population for safety analyses; the PP population was that for efficacy analyses. Experimental arms were compared individually to the control arm by ANCOVA. Analyses were adjusted for baseline differences in the parameter of interest. We considered adjustment for multiple baseline factors in each analysis, but found that the effect of additional factors was small. We did not adjust for multiple comparisons, but instead compared each candidate HDT

independently to the control arm, as the development of each candidate might be advanced independently based on study findings. The effects of treatment on CRP were examined using a random effects model for the repeated measurements of CRP on study day, and included an interaction between day and study arm, and the study arm slope parameter reported. CRP values were log-log transformed to improve linearity over time. Statistical analyses were performed using Stata (College Station, TX).

In accord with the data access provisions specified in the grant agreement, anonymized patient level data will be made available to qualified researchers through the London School of Hygiene and Tropical Medicine, at https://doi.org/10.17037/DATA.00002277.

## Results

A CONSORT diagram appears as Fig 1. Baseline subject characteristics appear in Table 1. Patients were generally young, predominantly male, with cavitary disease, heavy sputum infection burdens, and moderate impairment of lung function. Baseline parameters were reasonably balanced across study arms.

A representative transverse fused PET/CT image with its selected region of interest appears in Fig 2. Across all study arms, baseline values for mean log peak and maximum glycolytic activity were similar (1.01 and 1.12, respectively, Table 1, corresponding to numeric values of 10.2 and 13.2). The mean log total glycolytic activity (the log product of activity and volume) was 3.60, corresponding to a numeric value of 3981. The mean log total radiodensity using the modified Hounsfield scale was 3.05; for comparison, a 3000mL volume comprised of equal portions of air and water would show a log total radiodensity on this scale of 3.18.

Treatment effects on PET/CT parameters appear in Table 2. All four PET/CT parameters showed significant decreases from baseline to day 56 regardless of study treatment. The greatest and most consistent change (approximately a 30% reduction) occurred for total glycolytic activity (SUVbw*ml). The magnitude of the change in total radiodensity was only a third of that for total glycolytic activity; nonetheless, the reduction was highly significant, reflecting its reduced variability compared to PET. Of the four host-directed therapies, only everolimus showed a further reduction on day 56 compared to control. The log -0.117 treatment effect (a reduction of approximately 25% *vs* control), was evident in both peak and maximum glycolytic activity, but not in total glycolytic activity. None of the HDTs showed an effect on total radiodensity beyond that of anti-tuberculous treatment alone.

The mean CRP at baseline was 80.6 mg/L (Table 1). It dropped precipitously early during treatment but did so equally in all treatment arms (Table 3 and Fig 3). A trend toward a superior rate of reduction in CRP due to CC-11050 approached but did not reach statistical significance when analyzed according to the slope of log-log transformed values throughout treatment (Table 4).

Mean values of mycobactericidal activity in *ex vivo* whole blood culture for each treatment arm at 0–8 hr post-dose time points on day 84 appear in Fig 4. In the absence of TB chemotherapy, *M tuberculosis* H37Rv typically shows 0.2 log/day growth in whole blood culture [10]. As Fig 4 indicates, killing, rather than growth, is apparent throughout the dosing interval on day 84 in all study arms, attributable to the 45 hr plasma half-life of rifabutin [11]. Maximum antibacterial effects (*ie*, minimum mycobacterial viability) occurred 3–4 hrs post dose. Summary measures of mycobactericidal activity (maximal and total effects throughout the dosing interval) appear in Table 5. Only auranofin showed superior activity compared to control. This was most evident on day 84, and most evident in its maximum effect. The effect of auranofin was lost by day 140, 5 weeks after its discontinuation.

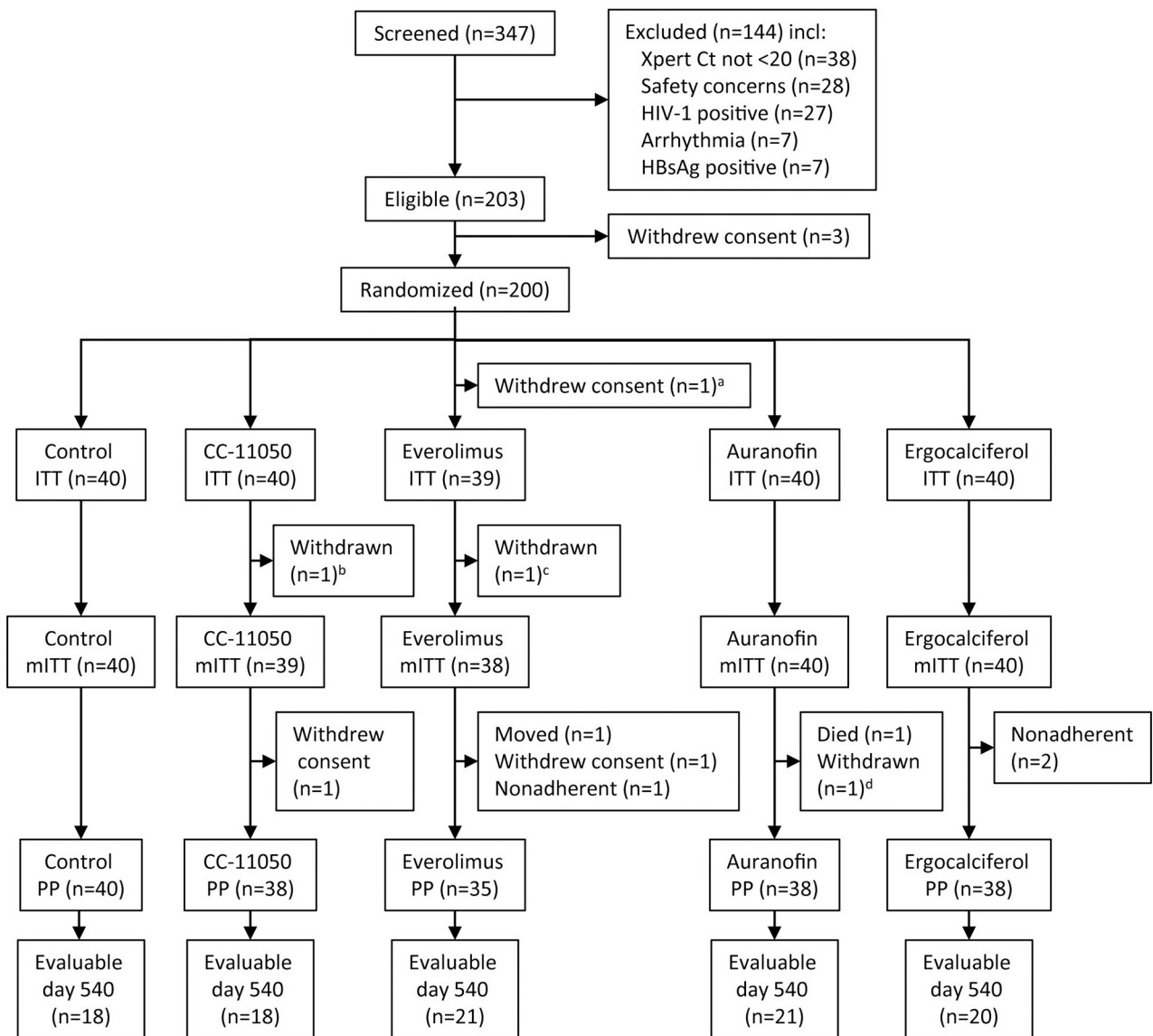

**Fig 1. CONSORT diagram.** [a]Withdrew consent prior to receiving any treatment. [b]Found to have spinal tuberculosis on baseline PET/CT scan. [c]Found to have elevated blood glucose at screening. [d]Required an excluded concomitant medication. ITT = intent to treat; PP = per protocol; Ct = cycle threshold. From [12].

## Discussion

This study examined the effects on 3 early biomarkers of four host-directed therapy candidates in an experimental medicine phase II clinical trial. As we have previously reported [12], the study found that CC-11050 and everolimus both enhanced the recovery of FEV1, a generalizable independent predictor of all-cause mortality [3]. In low-income countries where tuberculosis is most prevalent, standardized mortality risk doubles as FEV1 declines to 70% of predicted, and doubles again as it further declines to 50% [3]. This estimate, of a 2 to 4-fold increase in all-cause mortality post-TB due to permanent loss of FEV1, is consistent with findings of multiple retrospective studies showing excess post-TB deaths due to unexpected cardiovascular and respiratory illness [4, 5]. Interventions to protect the lung and reduce lung

**Table 1. Baseline characteristics of the per protocol population.**

|  | Control (N = 40) | CC-11050 (N = 38) | Everolimus (N = 35) | Auranofin (N = 38) | Ergocalciferol (N = 38) | Total (N = 189) |
|---|---|---|---|---|---|---|
| **Clinical** |  |  |  |  |  |  |
| Age years, median (IQR) | 32 (26;42.5) | 33.5 (27;39) | 32 (26;42) | 37.5 (30;44) | 38 (28;49) | 35 (27;42) |
| Weight kg, median (IQR) | 54.7 (51.8;61.3) | 54.5 (49.2;61.6) | 55.2 (51.9;59.7) | 53.4 (50.2;59.3) | 53.9 (49.8;59.5) | 54.4 (50.5;59.7) |
| BMI kg/m$^2$, median (IQR) | 18.5 (16.9;19.9) | 18.7 (16.8;19.7) | 19.4 (17.4;21.5) | 18.4 (17.3;20.0) | 18.3 (16.6;19.7) | 18.6 (17.1;20.1) |
| Female, N (%) | 2 (5.0) | 7 (18.4) | 4 (11.4) | 3 (7.89) | 7 (18.4) | 23 (12.2) |
| Smoking history, N (%) | 25 (62.5) | 22 (57.9) | 16 (45.7) | 19 (50.0) | 17 (44.7) | 99 (52.4) |
| **Radiography** |  |  |  |  |  |  |
| Total cavity diameter, N (%) |  |  |  |  |  |  |
| 0 cm | 11 (27.5) | 4 (10.5) | 7 (20.0) | 2 (5.26) | 3 (7.89) | 27 (14.3) |
| >0 and <4 cm | 24 (60.0) | 23 (60.5) | 18 (51.4) | 25 (65.8) | 27 (71.1) | 117 (61.9) |
| ≥4 cm | 5 (12.5) | 11 (29.0) | 10 (28.6) | 11 (29.0) | 8 (21.1) | 45 (23.8) |
| Radiographic extent of disease, N (%) |  |  |  |  |  |  |
| Moderately advanced | 24 (60.0) | 22 (57.9) | 23 (65.7) | 21 (55.3) | 20 (52.6) | 110 (58.2) |
| Far advanced | 16 (40.0) | 16 (42.1) | 12 (34.3) | 17 (44.7) | 18 (47.4) | 79 (41.8) |
| **Spirometry** |  |  |  |  |  |  |
| FEV1%, mean (95%CI) | 61.7 (56.3;67.1) | 62.3 (54.9;69.7) | 70.1 (63.7;76.5) | 60.1 (53.4;66.7) | 65.2 (56.8;73.6) | 63.8 (60.7;66.8) |
| FVC, L mean (95%CI) | 3.12 (2.88;3.37) | 2.96 (2.63;3.29) | 3.12 (2.83;3.41) | 2.97 (2.70;3.24) | 2.97 (2.61;3.33) | 3.03 (2.90;3.16) |
| **Microbiology** |  |  |  |  |  |  |
| Xpert Ct, mean (95%CI) | 16.4 (15.4;17.4) | 16.5 (15.4;17.6) | 17.7 (16.1;19.2) | 17.4 (16.3;18.6) | 16.3 (15.1;17.6) | 16.9 (16.3;17.4) |
| MGIT TTP hrs, median (IQR) | 122 (94.9;149) | 117 (103;132) | 116 (96.2;136) | 110 (96.1;123) | 115 (102;127) | 116 (108;124) |
| **PET/CT** |  |  |  |  |  |  |
| Log peak SUVbw, mean (95%CI) | 1.00 (0.95;1.06) | 1.00 (0.94;1.06) | 0.96 (0.88;1.03) | 1.04 (1.00;1.09) | 1.05 (0.98;1.12) | 1.01 (0.99;1.04) |
| Log maximum SUVbw, mean (95%CI) | 1.11 (1.06;1.16) | 1.11 (1.06;1.16) | 1.08 (1.02;1.14) | 1.14 (1.10;1.18) | 1.16 (1.10;1.21) | 1.12 (1.10;1.14) |
| Log total SUVbw*ml, mean (95%CI) | 3.62 (3.55;3.69) | 3.57 (3.51;3.63) | 3.52 (3.45;3.59) | 3.63 (3.57;3.70) | 3.63 (3.57;3.70) | 3.60 (3.57;3.63) |
| Log total mHU*ml, mean (95%CI) | 3.06 (3.03;3.10) | 3.03 (3.00;3.06) | 3.03 (2.99;3.06) | 3.63 (3.57;3.70) | 3.08 (3.05;3.11) | 3.05 (3.04;3.07) |
| **CRP mg/L (95%CI)** | 74.7 (60.7;88.7) | 76.6 (57.6;95.5) | 66.9 (44.1;89.7) | 93.2 (73.1;113.4) | 90.2 (69.4;111.0) | 80.6 (72.1;89.1) |

IQR = interquartile range; BMI = body mass index; CI = confidence interval; FEV1 = 1-second forced expiratory volume; FVC = forced vital capacity; Xpert Ct = cycle threshold; MGIT TTP = time to positivity in mycobacterial growth indicator tube cultures; PET = positron emission tomography; CT = computer tomography; SUVbw = standardized uptake value adjusted for body weight; mHU = modified Hounsfield units; CRP = C reactive protein.

inflammation may potentially offset as much as half of this excess mortality risk. However, FEV1 appears at best to be an intermediate rather than early biomarker of lung protection in tuberculosis. Although trends were identified at early time points for FEV1 in the trial, they did not reach statistical significance until the end of tuberculosis treatment on day 180. The finding limits the utility of FEV1 as an early marker of HDT effects, and underscores the need for alternative early TB-HDT biomarkers.

Perhaps the most striking observation of this report therefore is the apparent inability of multiple early markers of inflammation and infection to broadly predict long-term TB-HDT effects. Three of the markers–blood CRP, total lung radiodensity as measured by CT, and total lung glycolytic activity as measured by [18]F-FDG PET–showed highly significant early responses to antimicrobial chemotherapy, yet none provided early evidence that either of two anti-inflammatory HDTs ultimately would protect the lung. The two remaining markers– maximal or peak glycolytic activity, and whole blood bactericidal activity–yielded inconsistent findings across arms when compared to older biomarkers with greater accumulated

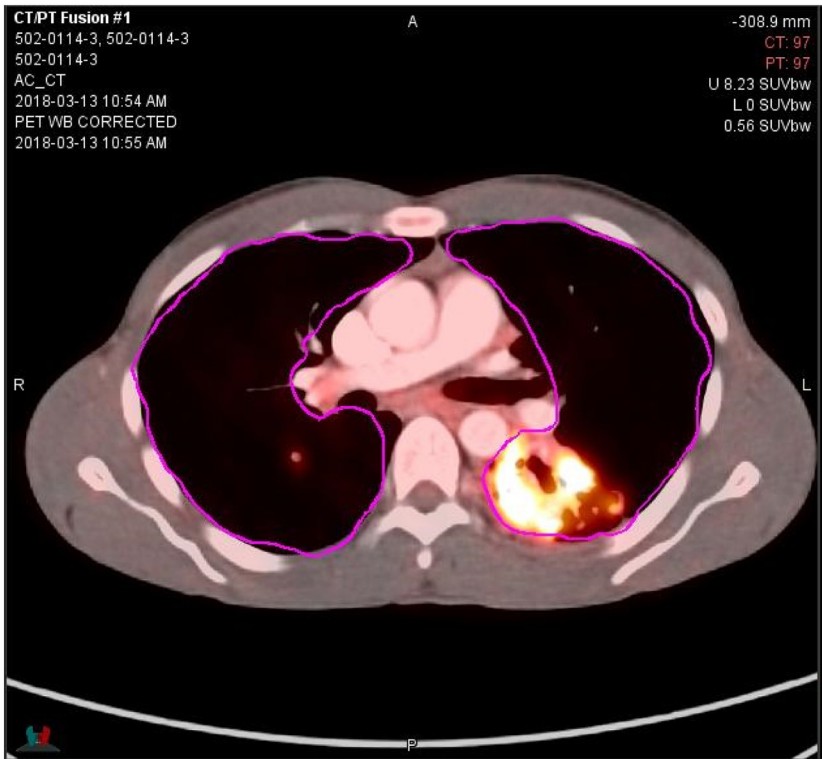

**Fig 2. Representative PET/CT fused image.** Gray scale indicates CT radiodensity; colored scale indicates PET glycolytic activity. Pink lines indicate limits of selected region of interest.

experience. This appears to indicate that distinct biomarkers may be required to assess early effects of TB-HDTs.

Glycolytic activity detected by [18]F-FDG PET reflects glucose utilization by mammalian cells. There is considerable interest in PET/CT as a biomarker of TB treatment at time points as early as 14 days [13]. The ability of PET/CT to predict clinical tuberculosis outcomes remains uncertain. Although one study in patients with drug-resistant tuberculosis found changes in PET signal at 2 months predicted treatment outcome [14], another in patients with drug-susceptible disease found persisting PET abnormalities at the end of apparently successful treatment in a majority of patients [15]. We hypothesized that as a measure of lung inflammation, early decreases in PET signal would generally be associated with superior recovery of lung function. Instead, we found divergent PET responses to everolimus and CC-11050, despite nearly identical effects on FEV1 [12]. The particular responsiveness of [18]F-FDG PET to everolimus may reflect the close linkage of mTOR activation specifically to glucose utilization rather than to inflammation more generally [16]. This appears to indicate that distinct early biomarkers for host-directed therapies may be required according to their mechanism of action. We also found that the effect of everolimus was most evident in measures of peak or maximum glycolytic activity. This may indicate that it is most active in highly inflammatory lesions.

CC-11050 previously was the backup compound for apremilast, a phosphodiesterase inhibitor now approved for multiple anti-inflammatory indications. Although CC-11050 had no effect on early PET/CT signals, we did observe a trend toward reduced CRP throughout treatment. This became apparent only after log-log transformation of CRP values, which improved

**Table 2. Effect of treatment on PET/CT parameters.**

| | Change from baseline to day 56 | | | Difference from control at day 56 | | | | | |
| --- | --- | --- | --- | --- | --- | --- | --- | --- | --- |
| | | | | Unadjusted | | | Adjusted for differences at baseline | | |
| | Mean | 95% CI | P | Mean | 95% CI | P | Mean | 95% CI | P |
| **Log peak SUVbw** | | | | | | | | | |
| Control | -0.089 | -0.131; -0.042 | <0.001 | 0 | - | - | 0 | - | - |
| CC-11050 | -0.087 | -0.152; -0.022 | 0.010 | -0.010 | -0.098; 0.078 | 0.824 | -0.006 | -0.085; 0.073 | 0.888 |
| Everolimus | -0.178 | -0.253; -0.092 | <0.001 | -0.140 | -0.229; -0.050 | 0.002 | -0.117 | -0.197; -0.037 | 0.005 |
| Auranofin | -0.101 | -0.169; -0.038 | 0.003 | 0.018 | -0.069; 0.106 | 0.684 | 0.001 | -0.078; 0.079 | 0.986 |
| Ergocalciferol | -0.133 | -0.203; -0.076 | <0.001 | -0.014 | -0.102; 0.074 | 0.706 | -0.029 | -0.108; 0.050 | 0.466 |
| **Log max SUVbw** | | | | | | | | | |
| Control | -0.060 | -0.101; -0.019 | 0.006 | 0 | - | - | 0 | - | - |
| CC-11050 | -0.070 | -0.130; -0.009 | 0.025 | -0.014 | -0.085; 0.056 | 0.685 | -0.013 | -0.079; 0.054 | 0.704 |
| Everolimus | -0.134 | -0.201; -0.067 | <0.001 | -0.111 | -0.182; -0.040 | 0.002 | -0.100 | -0.168; -0.033 | 0.004 |
| Auranofin | -0.077 | -0.139; -0.015 | 0.016 | 0.002 | -0.068; 0.072 | 0.958 | -0.007 | -0.073; 0.059 | 0.844 |
| Ergocalciferol | -0.133 | -0.190; -0.075 | <0.001 | -0.039 | -0.109; 0.031 | 0.274 | -0.050 | -0.117; 0.016 | 0.137 |
| **Log total SUVbw*ml** | | | | | | | | | |
| Control | -0.154 | -0.190; -0.119 | <0.001 | 0 | - | - | 0 | - | - |
| CC-11050 | -0.144 | -0.187; -0.100 | <0.001 | -0.045 | -0.130; 0.040 | 0.296 | -0.006 | -0.061; 0.050 | 0.838 |
| Everolimus | -0.159 | -0.201; -0.117 | <0.001 | -0.112 | -0.198; -0.025 | 0.012 | -0.038 | -0.095; 0.019 | 0.184 |
| Auranofin | -0.158 | -0.203; -0.112 | <0.001 | 0.004 | -0.081; 0.088 | 0.934 | -0.003 | -0.058; 0.053 | 0.928 |
| Ergocalciferol | -0.180 | -0.234; -0.125 | <0.001 | -0.023 | -0.108; 0.062 | 0.598 | -0.018 | -0.074; 0.037 | 0.519 |
| **Log total mHU*ml** | | | | | | | | | |
| Control | -0.050 | -0.065; -0.036 | <0.001 | 0 | - | - | 0 | - | - |
| CC-11050 | -0.040 | -0.057; -0.023 | <0.001 | -0.025 | -0.069; 0.018 | 0.252 | 0.005 | -0.018; 0.027 | 0.692 |
| Everolimus | -0.059 | -0.078; -0.041 | <0.001 | -0.049 | -0.093; -0.005 | 0.030 | -0.016 | -0.039; 0.007 | 0.160 |
| Auranofin | -0.045 | -0.060; -0.031 | <0.001 | 0.015 | -0.028; 0.058 | 0.501 | 0.006 | -0.016; 0.028 | 0.610 |
| Ergocalciferol | -0.064 | -0.084; -0.044 | <0.001 | -0.015 | -0.058; 0.029 | 0.512 | -0.011 | -0.034; 0.011 | 0.328 |

In the left panel, differences were determined by paired t test. In the right panel, differences were determined by ANCOVA, before and after adjustment based on differences from control at baseline in that parameter (*eg*, analysis of log peak SUVbw was adjusted for baseline differences in log peak SUVbw). SUVbw = standardized uptake value adjusted for body weight. CI = confidence interval; mHU = modified Hounsfield units.

**Table 3. Early treatment effects on CRP.**

| | Change from baseline to day 56 | | | Difference from control at day 56 | | | | | |
| --- | --- | --- | --- | --- | --- | --- | --- | --- | --- |
| | | | | Unadjusted | | | Adjusted | | |
| | Mean | 95% CI | P | Mean | 95% CI | P | Mean | 95% CI | P |
| Control | -57.42 | -71.86; -42.98 | <0.0001 | 0 | - | - | 0 | - | - |
| CC-11050 | -62.13 | -81.40; -42.86 | <0.0001 | -5.09 | -28.47; 18.30 | 0.668 | -3.53 | -11.75; 4.69 | 0.398 |
| Everolimus | -44.65 | -69.54; -20.57 | <0.0001 | 12.39 | -11.68; 36.46 | 0.311 | 5.96 | -2.51; 14.42 | 0.167 |
| Auranofin | -73.11 | -94.13; -52.09 | <0.0001 | -16.07 | -39.45; 7.32 | 0.177 | -0.66 | -8.92; 7.59 | 0.874 |
| Ergocalciferol | -73.36 | -94.64; -51.58 | <0.0001 | -16.32 | -39.87; 7.22 | 0.173 | -0.3.21 | -11.52; 5.09 | 0.446 |

In the left panel, differences were determined by paired t test. In the right panel, differences were determined by ANCOVA, before and after adjustment based on differences from control at baseline in CRP. CI = confidence interval.

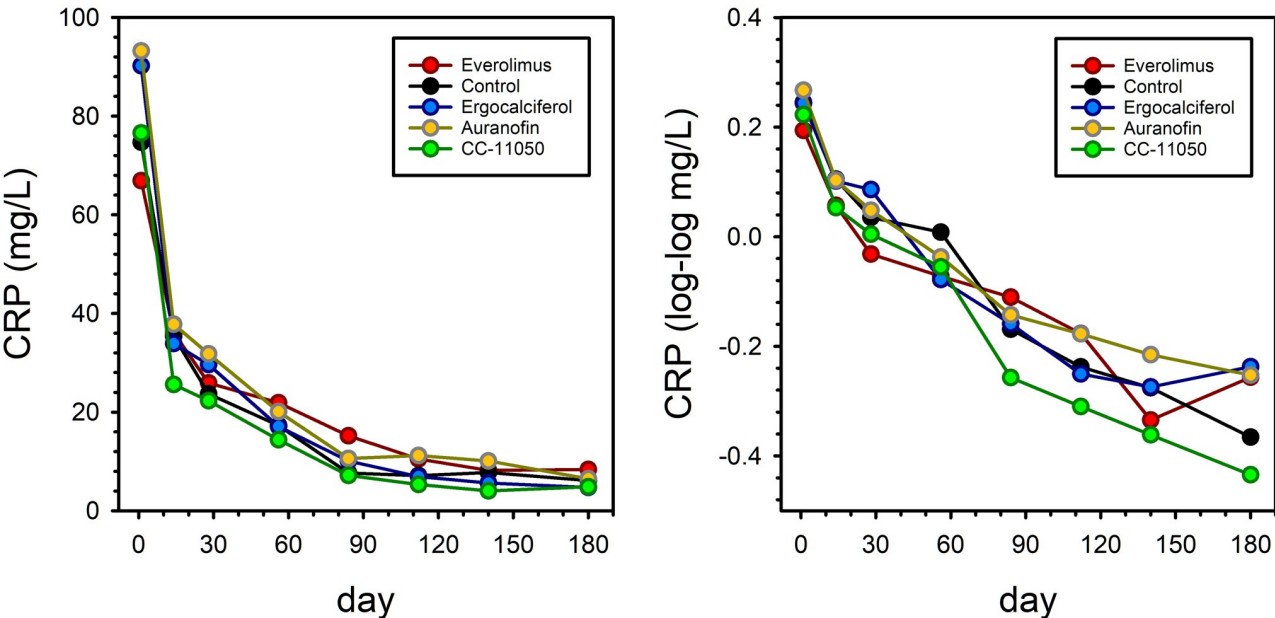

**Fig 3. Treatment effects on CRP before and after log-log transformation.** Symbols indicate mean values.

linearity over time and increased relative differences at low values. CRP values remained elevated in all arms; on day 180, means were 4.7–8.4 mg/L, well above the level of 3.0 associated with cardiovascular risk [17, 18]. Studies of long-term post-TB cardiovascular risks in relation to CRP at the conclusion of treatment appear warranted.

Auranofin was introduced in the 1980s as a disease-modifying treatment for rheumatoid arthritis. Gold salts show broad-spectrum antibacterial and antiviral activity *in vitro* that includes such diverse pathogens as *M tuberculosis* and SARS-CoV-2 [19, 20]. In past centuries, sanocrysin, a double thiosulphate of gold(III) and sodium, was widely used for tuberculosis treatment [21]. Its use was largely abandoned in 1931 when a careful clinical trial showed toxicity without apparent clinical benefit [22]. Gold becomes highly bound to macrophage and serum proteins *in vivo* [23], with uncertain effects on its tissue distribution and antimicrobial activity. As such, gold salts may be considered dependent on accumulation in host cells for its action rather than host-directed *per se*. Auranofin was the only TB-HDT to augment intracellular mycobactericidal activity in *ex vivo* whole blood cultures in this study. Mycobacteria added to whole blood cultures are rapidly ingested by phagocytic cells [24] and are subject to

**Table 4. Effects on log-log CRP slope throughout treatment.**

| | Mean | Difference from control | | | Adjusted[a] difference from control | | |
|---|---|---|---|---|---|---|---|
| | | Mean | 95% CI | P | Mean | 95% CI | P |
| Control | -0.00393 | 0 | - | - | 0 | - | - |
| CC-11050 | -0.00475 | -0.00073 | -0.00168; 0.00022 | 0.133 | -0.00082 | -0. 00177; 0.00014 | 0.094 |
| Everolimus | -0.00312 | 0.00090 | -0.00008; 0.00188 | 0.070 | 0.00083 | -0. 00016; 0.00181 | 0.100 |
| Auranofin | -0.00382 | 0.00020 | -0.00074; 0.00114 | 0.675 | 0.00012 | -0. 00083; 0.00107 | 0.809 |
| Ergocalciferol | -0.00438 | -0.00036 | -0.00131; 0.00060 | 0.464 | -0.00044 | -0. 00140; 0.00052 | 0.366 |

[a]Values are adjusted for baseline differences in log-log CRP. CI = confidence interval. A negative mean difference from control indicates a more rapid decline.

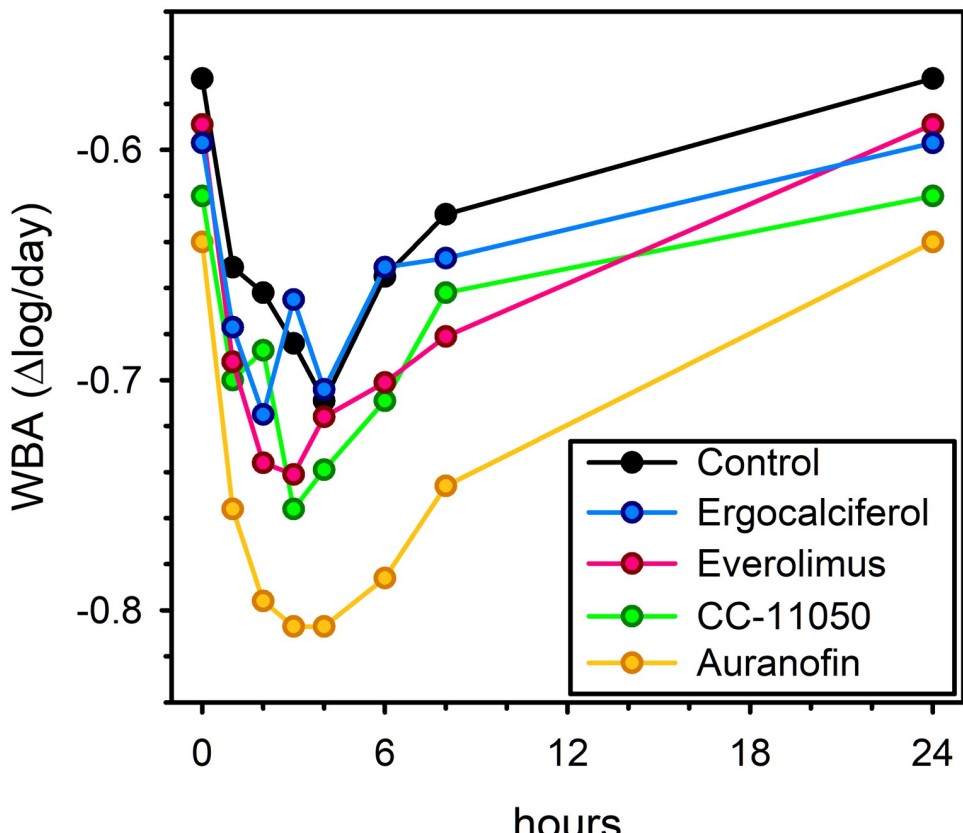

**Fig 4. Mean whole blood bactericidal activity (WBA) by treatment arm, on study day 84, prior to and at intervals after dosing.** The curve for CC-11050 does not include a potential effect of the second daily drug dose. Negative values indicate killing.

**Table 5. Whole blood bactericidal activity (WBA).**

| | Day 42 | | | | Day 84 | | | | Day 140 | | | |
|---|---|---|---|---|---|---|---|---|---|---|---|---|
| | Mean | Difference from control | | | Mean | Difference from control | | | Mean | Difference from control | | |
| | | Mean | 95% CI | P | | Mean | 95% CI | P | | Mean | 95% CI | P |
| **Total effect per dose (Δlog)** | | | | | | | | | | | | |
| Control | -0.627 | 0 | - | - | -0.619 | 0 | - | - | -0.642 | 0 | - | - |
| CC-11050 | -0.669 | -0.042 | -0.154; 0.069 | 0.452 | -0.662 | -0.043 | -0.148; 0.062 | 0.415 | -0.571 | 0.072 | -0.024; 0.167 | 0.137 |
| Everolimus | -0.626 | 0.002 | -0.112; 0.116 | 0.979 | -0.657 | -0.038 | -0.130; 0.054 | 0.406 | -0.658 | -0.015 | -0.108; 0.078 | 0.743 |
| Auranofin | -0.703 | -0.075 | -0.186; 0.035 | 0.178 | -0.721 | -0.102 | -0.198; -0.005 | 0.039 | -0.622 | 0.021 | -0.084; 0.125 | 0.691 |
| Ergocalciferol | -0.661 | -0.033 | -0.132; 0.065 | 0.498 | -0.638 | -0.019 | -0.123; 0.085 | 0.715 | -0.696 | -0.054 | -0.137; 0.029 | 0.199 |
| **Maximal effect (Δlog /day)** | | | | | | | | | | | | |
| Control | -0.755 | 0 | - | - | -0.767 | 0 | - | - | -0.788 | 0 | - | - |
| CC-11050 | -0.820 | -0.065 | -0.173; 0.042 | 0.229 | -0.817 | -0.050 | -0.158; 0.058 | 0.354 | -0.758 | 0.030 | -0.072; 0.131 | 0.562 |
| Everolimus | -0.815 | -0.060 | -0.169; 0.048 | 0.269 | -0.837 | -0.070 | -0.153; 0.013 | 0.095 | -0.829 | -0.041 | -0.145; 0.063 | 0.431 |
| Auranofin | -0.892 | -0.137 | -0.256; -0.017 | 0.026 | -0.896 | -0.129 | -0.223; -0.034 | 0.009 | -0.790 | -0.002 | -0.109; 0.105 | 0.972 |
| Ergocalciferol | -0.816 | -0.061 | -0.168; 0.046 | 0.256 | -0.804 | -0.038 | -0.152; 0.077 | 0.509 | -0.857 | -0.069 | -0.157; 0.019 | 0.121 |

*Negative values indicate killing. Differences from control were determined by t test. CI = confidence interval. Values for CC-11050 may underestimate total effects as the period of sampling (0–8 hrs) did not include the evening dose.

the combined effects of cellular immunity and administered treatments [25]. The finding that nearly 3 months of daily auranofin dosing was required to reach the maximum mycobactericidal effect of auranofin is consistent with its high level of protein binding and large apparent plasma volume of distribution. At month-3, auranofin added log 0.129 (about 33%) to the intracellular mycobactericidal activity of rifabutin plus isoniazid. However, auranofin failed to effect sputum culture conversion in this study (HR = 1.17, 95%CI = 0.74 to 1.84, P = 0.51) [12]. The most likely explanation may be that auranofin accumulates and exerts antimycobacterial activity in blood phagocytic cells, but that these cells have limited ability to penetrate mature lung granulomas to deliver an antimicrobial payload. If cellular lesional penetration is indeed the limiting factor, it may be equally problematic for other TB-HDTs such as imatinib that depend on migration of activated cells from the bone marrow to the site of infection.

The main strengths of this study are the diversity of the therapeutic interventions and biomarker endpoints. However, several limitations should be considered. The trial's experimental medicine design limited its sample size and prevented the blinded use of placebos. These factors limited statistical power and potentially introduced bias. Key findings will require verification in future trials. For the PET/CT scans, our use of ROIs without prior selection according to lung density may have reduced our ability to detect small changes in total glycolytic activity. Larger studies with longer follow-up will be necessary to directly assess effects on mortality.

In summary, in this experimental medicine study of 4 TB-HDTs, early biomarker detection of treatment effects appeared dependent on the specific mechanism of action of each therapy candidate.

## Acknowledgments

The authors would like to thank Andrew Nunn (British Medical Research Council, now University College London), Michael Hoelscher (University of Munich) and Andreas Diacon (Stellenbosch University) for serving as members of the Data and Safety Monitoring Committee. Some of this material was presented as a late-breaking abstract at the 2019 annual meeting of the American Thoracic Society.

## Author Contributions

**Conceptualization:** Robert S. Wallis.

**Formal analysis:** Robert S. Wallis, Sibuse Ginindza, Katherine Fielding.

**Funding acquisition:** Robert S. Wallis.

**Investigation:** Robert S. Wallis, Nishanee Arjun, Morongwe Likoti, Modulakgotla Sebe, Mohammed Rassool, Khatija Ahmed, Bintou A. Ahidjo, Mboyo D. T. Vangu.

**Methodology:** Robert S. Wallis.

**Project administration:** Robert S. Wallis, Trevor Beattie, Nishanee Arjun, Morongwe Likoti, Modulakgotla Sebe, Vinodh A. Edward, Mohammed Rassool, Khatija Ahmed, Gavin Churchyard.

**Writing – original draft:** Robert S. Wallis.

**Writing – review & editing:** Robert S. Wallis, Sibuse Ginindza, Trevor Beattie, Nishanee Arjun, Morongwe Likoti, Vinodh A. Edward, Mohammed Rassool, Khatija Ahmed, Katherine Fielding, Bintou A. Ahidjo, Mboyo D. T. Vangu, Gavin Churchyard.

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
