## [Decision Letter · Decision Letter 0]

15 Apr 2021

PONE-D-21-03831

Lung and blood early biomarkers for host-directed tuberculosis therapies: Secondary outcomes from a randomized controlled trial

PLOS ONE

Dear Dr. Wallis,

Thank you for submitting your manuscript to PLOS ONE. After careful consideration, we feel that it has merit but does not fully meet PLOS ONE’s publication criteria as it currently stands. Therefore, we invite you to submit a revised version of the manuscript that addresses the points raised during the review process and you will find below.

We look forward to receiving your revised manuscript.

Kind regards,

Christophe Leroyer

Academic Editor

PLOS ONE

Journal Requirements:

**Comments to the Author**

1. Is the manuscript technically sound, and do the data support the conclusions?

Reviewer #1: Yes

Reviewer #2: Yes

2. Has the statistical analysis been performed appropriately and rigorously? 

Reviewer #1: Yes

Reviewer #2: Yes

3. Have the authors made all data underlying the findings in their manuscript fully available?

Reviewer #1: Yes

Reviewer #2: Yes

4. Is the manuscript presented in an intelligible fashion and written in standard English?

Reviewer #1: Yes

Reviewer #2: Yes

5. Review Comments to the Author

Reviewer #1: For Tables 2 - 5, please include the statistical analysis method used to calculate the CI's and p-values in either the description of the table or in the table footnote.

The text in the statistical analysis section mentions that ANCOVA was used and that analyses were adjusted for the relevant baseline characteristics. Please indicate what these characteristics are. It is not appropriate to leave it up to the reader to guess them.

Reviewer #2: The persistance of tuberculosis damage on the lung has a great impact on patients' quality of life and prognosis. The final aim of the work was to identify host directed drugs that would be able to improve recovery of lung function. The authors published a first paper using FEV1 as a surrogate of HTD efficacy. It worked, but the impact wasn't demonstrable before the end of the treatment. The aim of the second paper was to select biomarkers that would reflect early improvement due to the tested drugs. Several positive results are reported. However, it seems difficult to evaluate in the same experiment the drugs and the biomarkers. If the drugs are not working well, how could the authors identify the best biomarker? This point could be discussed in the discussion.

Some minor points could be modified:

the results about FEV1 are already published and reported in the introduction. Why reporting them in the results part of the abstract?

auranofin has a direct microbicidal activity as the authors reported. Could this property explain the enhancement of mycobacteria killing?

The trend of its effect on CRP is , as the authors say, not significant. Has it to be considered?

6. PLOS authors have the option to publish the peer review history of their article (what does this mean?). If published, this will include your full peer review and any attached files.

Reviewer #1: No

Reviewer #2: No

---

## [Author Response · Author response to Decision Letter 0]

28 Apr 2021

27 April 2021

Re: PONE-D-21-03831

Thank you very much for the opportunity to revise this manuscript. We have addressed the questions as follows: 

Editor’s comments:

If applicable, we recommend that you deposit your laboratory protocols in protocols.io to enhance the reproducibility of your results. 

We have placed the protocol online and have included a link to its URL in the text. 

2. In your Data Availability statement, you have not specified where the minimal data set underlying the results described in your manuscript can be found. 

We are currently working with the Critical Path Institute to place the full study dataset online as a part of the ERA4TB Consortium. We would like to include the full PET/CT images. This has proven challenging, as the scans are about 1TB in total. The text has been modified to indicate this. 

Reviewers’ comments:

Reviewer #1: For Tables 2 - 5, please include the statistical analysis method used to calculate the CI's and p-values in either the description of the table or in the table footnote.

This has been added. 

The text in the statistical analysis section mentions that ANCOVA was used and that analyses were adjusted for the relevant baseline characteristics. Please indicate what these characteristics are. It is not appropriate to leave it up to the reader to guess them.

This has been added. 

Reviewer #2: The persistance of tuberculosis damage on the lung has a great impact on patients' quality of life and prognosis. The final aim of the work was to identify host directed drugs that would be able to improve recovery of lung function. The authors published a first paper using FEV1 as a surrogate of HTD efficacy. It worked, but the impact wasn't demonstrable before the end of the treatment. The aim of the second paper was to select biomarkers that would reflect early improvement due to the tested drugs. Several positive results are reported. However, it seems difficult to evaluate in the same experiment the drugs and the biomarkers. If the drugs are not working well, how could the authors identify the best biomarker? This point could be discussed in the discussion.

It is generally thought that the most valuable biomarkers are agnostic as to the mechanism of the intervention. We did not identify any biomarkers meeting this standard. The text has been revised to reflect this. 

Some minor points could be modified:

the results about FEV1 are already published and reported in the introduction. Why reporting them in the results part of the abstract?

We have edited the abstract to keep this to a minimum. 

Auranofin has a direct microbicidal activity as the authors reported. Could this property explain the enhancement of mycobacteria killing?

Yes, we believe this is the case. The text has been revised to clarify this point. 

The trend of its effect on CRP is , as the authors say, not significant. Has it to be considered?

Yes, we have edited the text to emphasize the effect did not reach statistical significance.

---

## [Editor Report · Decision Letter 1]

10 May 2021

Lung and blood early biomarkers for host-directed tuberculosis therapies: Secondary outcomes from a randomized controlled trial

PONE-D-21-03831R1

Dear Dr. Wallis,

We’re pleased to inform you that your manuscript has been judged scientifically suitable for publication and will be formally accepted for publication once it meets all outstanding technical requirements.

Kind regards,

Christophe Leroyer

Academic Editor

PLOS ONE

---

## [Editor Report · Acceptance letter]

27 Jan 2022

PONE-D-21-03831R1 

Lung and blood early biomarkers for host-directed tuberculosis therapies: Secondary outcome measures from a randomized controlled trial 

Dear Dr. Wallis:

I'm pleased to inform you that your manuscript has been deemed suitable for publication in PLOS ONE. Congratulations! Your manuscript is now with our production department. 

Kind regards, 

on behalf of

Dr. Christophe Leroyer 

Academic Editor

PLOS ONE